# Multi-Level Resistive Al/Ga_2_O_3_/ITO Switching Devices with Interlayers of Graphene Oxide for Neuromorphic Computing

**DOI:** 10.3390/nano13121851

**Published:** 2023-06-13

**Authors:** Li-Wen Wang, Chih-Wei Huang, Ke-Jing Lee, Sheng-Yuan Chu, Yeong-Her Wang

**Affiliations:** 1Department of Electrical Engineering, National Cheng-Kung University, Tainan 701, Taiwan; jk220052@gmail.com (L.-W.W.); chusy@mail.ncku.edu.tw (S.-Y.C.); 2Institute of Microelectronics, Department of Electrical Engineering, National Cheng-Kung University, Tainan 701, Taiwan; william9450504@gmail.com; 3Program on Semiconductor Process Technology, Academy of Innovative Semiconductor and Sustainable Manufacturing, National Cheng-Kung University, Tainan 701, Taiwan

**Keywords:** RRAM, gallium oxide, graphene oxide, bilayer structure, multi-level storage

## Abstract

Recently, resistive random access memory (RRAM) has been an outstanding candidate among various emerging nonvolatile memories for high-density storage and in-memory computing applications. However, traditional RRAM, which accommodates two states depending on applied voltage, cannot meet the high density requirement in the era of big data. Many research groups have demonstrated that RRAM possesses the potential for multi-level cells, which would overcome demands related to mass storage. Among numerous semiconductor materials, gallium oxide (a fourth-generation semiconductor material) is applied in the fields of optoelectronics, high-power resistive switching devices, and so on, due to its excellent transparent material properties and wide bandgap. In this study, we successfully demonstrate that Al/graphene oxide (GO)/Ga_2_O_3_/ITO RRAM has the potential to achieve two-bit storage. Compared to its single-layer counterpart, the bilayer structure has excellent electrical properties and stable reliability. The endurance characteristics could be enhanced above 100 switching cycles with an ON/OFF ratio of over 10^3^. Moreover, the filament models are also described in this thesis to clarify the transport mechanisms.

## 1. Introduction

In the 20th century, D. Kahng and S. M. Sze proposed a floating gate (FG) device to obtain the non-volatility characteristic in memory applications. This was the first time that the possibility of nonvolatile MOS memory was recognized. It is worth mentioning that flash memory, based on MOSFET transistors with a floating gate, has become a common component of USB flash drives and solid-state disks (SSDs) [1].

Due to the physical limit, DRAM, SRAM, and flash memory are currently facing challenges in scaling down to 10 nm nodes. It has become much more difficult and expensive to enhance performance and reliability. At first, silicon-oxide-nitride-oxide-silicon (SONOS) flash memory, which uses a silicon nitride as a charge-trapping layer, relieved the problems caused by scaling down. However, when gate oxide thickness was reduced to about 7 nm, scaling down problems became inevitable [2].

RRAM, comprised of a simple capacitor-like structure, has great potential to replace flash memory and DRAM, becoming the next-generation nonvolatile memory. A great number of kinds of materials have been researched for application as a resistive switching layer, such as binary metal oxides, perovskite oxides, and chalcogenide materials. Above them, binary metal oxides such as HfO_x_, TiO_x_, AlO_x_, and ZrO_x_ have been identified as great candidates for use in switching layers owing to the simplicity of materials and their good compatibility with silicon CMOS fabrication. Thus, many research activities have bloomed, resulting in a few demonstrations of various RRAM structures with insulator layers comprising various binary metal oxides [3,4]. Wide bandgap materials, especially HfO_2_ (around 5.7 eV), play important roles in many fields of the semiconductor industry, such as MOSFET, RRAM, and laser diode fabrication. Ga_2_O_3_, which is thought of as a fourth-generation semiconductor material, has been demonstrated to be a promising material as a replacement for GaN and SiC because of its wide bandgap of 4.9 eV in the β-phase and its high breakdown field. Therefore, it is considered as the most appropriate candidate for high-power devices. However, some teams have studied Ga_2_O_3_ in memory applications. Based on the quantum effect, amorphous phase Ga_2_O_3_ (a-Ga_2_O_3_) is more suitable than β-phase Ga_2_O_3_ (β-Ga_2_O_3_) as the switching layer in RRAM because the leakage current is smaller and the bandgap of a-Ga_2_O_3_ is quite large [5]. In addition, it has inherently high resistance and very sensitive conductivity to oxygen, which leads to a high ON/OFF ratio.

According to previous studies, in order to improve the performance of RRAM in terms of gradient, films with oxygen content in metal oxides play a very important role [3,4]. The employed thin ZnO films grown by radio-frequency magnetron sputtering act as buffer layers that provide adequate oxygen vacancies while improving the performance of GZO RRAM [6]. Embedding a thin ZnO buffer layer in the ZnLaO RRAM using chemical solution deposition provides a simple path to the formation of oxygen vacancy filaments, thus improving RS performance [7]. However, the challenge presented by poor endurance must be addressed. The applied voltage forms conductive filaments in the oxygen vacancies in the switching layer; in other words, the most important oxygen vacancy in the resistive switch has a special role. Improved RS uniformity in GO-based RRAM devices was realized through an adjustment to reduce GO-domains using the photocatalytic reduction method. Thus, the formation locations and microstructures of the defective oxygen vacancy CFs were effectively confined [8]. The appearance of Al metal nanofilaments and evolution of oxygen content during the RS process was observed in Al/GO/Al memory devices, which indicates that the RS mechanism is attributed to the coordinated effect of the AlOx and GO layers. A low-voltage spherical aberration-corrected transmission electron microscopy was used to demonstrate the nanoscale conductive graphitic channels within graphene oxide films stemming from the detachment of oxygen functional groups (i.e., GO reduction) [9]. In order to improve durability, the GO film can act as an oxygen ion reservoir, with the switching behavior thus becoming more stable with the help of the O^2−^ storage layer caused by the double-layer structure.

In this study, bilayer Al/GO/a-Ga_2_O_3_/ITO RRAM was proposed and monolayer Al/a-Ga_2_O_3_/ITO RRAM was also fabricated to draw a comparison between the two structures. Compared with the monolayer structure, the bilayer one had better electrical properties and reliability characteristics. The multi-level characteristic was presented through modulation of current compliance, and the possibility of lower power consumption was also demonstrated due to lower operational voltage and current compliance.

## 2. Results and Discussion

Figure 1 shows the element distribution analyzed through energy dispersive spectroscopy (EDS) mapping. Based on this mapping, the relative positions of all elements can be clearly distinguished. The EDS mapping can also prove the scavenging phenomenon between the Al top electrode and the GO layer. Crystal structure analysis of Ga_2_O_3_ thin film on a glass substrate is depicted in Figure 2a. This result shows that the structure of Ga_2_O_3_ deposited by RF sputtering was in the amorphous phase. For the purpose of the low leakage current in RRAM devices, the amorphous phase is better than the crystalline phase [10]. It is noteworthy that the peak around 24 degrees was attributed to the substrate [11]. The bandgap of Ga_2_O_3_ varies in different phases [12,13]. In this measurement, the bandgap of the Ga_2_O_3_ film was confirmed by a U-3310 spectrophotometer, as shown in Figure 2b. The bandgap of Ga_2_O_3_ in the amorphous phase is around 4.25 eV. Raman spectroscopy is a useful and non-destructive method for exploring 2D materials. The Raman analysis is presented in Figure 3. The Raman spectrum of GO reveals three peaks at 1343, 1448, and 1601 cm^−1^. The peaks at 1343 and 1601 cm^−1^ indicate the D band and G band, respectively [14]. The peak at 1448 cm^−1^ is caused by C-C bonds, which are generated from the environment.

The Atomic Force Microscopy (AFM) images in Figure 4a,b reveal the surface morphology of the Ga_2_O_3_ and the Go/Ga_2_O_3_ thin films. The root-mean-square roughness (R_rms_) of theGa_2_O_3_ and the Go/Ga_2_O_3_ thin film was approximately 3.54 and 1.18 nm. Our previous sol–gel process research indicates that the reduction in the surface roughness of the thin film switching layer due to the insertion of a Go layer may affect the resistance switching characteristics.

The electrical properties of monolayer and bilayer structures were characterized by an Agilent B1500A parameter analyzer. The ITO bottom electrode was grounded, and the applied voltage was exerted on the Al top electrode. Figure 5 shows the resistive switching behaviors of the Al/Ga_2_O_3_/ITO/glass RRAM device and Al/GO/Ga_2_O_3_/ITO/glass RRAM device. The forming processes of the Al/Ga_2_O_3_/ITO/glass and Al/GO/Ga_2_O_3_/ITO/glass RRAM devices occurred under negative voltage sweeps (0 V → −10 V → 0 V) at −50 mV intervals, and the compliance current was set to 10 mA to prevent the soft breakdown of devices. In Figure 5a, the forming voltage of the Al/Ga_2_O_3_/ITO/glass RRAM device and Al/GO/Ga_2_O_3_/ITO/glass RRAM device was −7.3 V and −5.2 V, respectively. Figure 5b shows the I-V curve of the SET and RESET processes of the device. Obviously, the I-V curves exhibited bipolar switching behavior in both structures. The SET/RESET processes occurred under applied negative voltage sweeps (0 V → −4 V → 0 V) and positive voltage sweeps (0 V → +4 V → 0 V), respectively, and the compliance current was set to 100 mA. For monolayer structure RRAM, the set voltage and reset voltage were −1.75 V and 3.2 V, respectively. The ON/OFF ratio was approximately 10^3^ at −0.1 volts. The V_set_ and V_reset_ of Al/GO/Ga_2_O_3_/ITO/glass was −0.7 V and 2.2 V, respectively. The ON/OFF ratio was approximately 10^3^ at −0.1 volts. The ON/OFF ratios of the Al/Ga_2_O_3_/ITO/glass and Al/GO/Ga_2_O_3_/ITO/glass RRAM devices remained stable for more than 20 cycles.

So as to investigate the conduction mechanism of Al/Ga_2_O_3_/ITO/glass and Al/GO/Ga_2_O_3_/ITO/glass RRAM in the LRS and HRS, the DC I-V characteristics were fitted after we programed the device. Figure 4 shows the fitting results of the I-V characteristics. According to the Ohmic conduction equation (V = IR), current conduction is dominated by Ohmic conduction if the slope of close to 1. For Al/Ga_2_O_3_/ITO/glass and Al/GO/Ga_2_O_3_/ITO/glass structures, the slope of the LRS in the low-voltage region is around 1.00, which indicates that the current conduction behavior is Ohmic, as shown in Figure 6a,b. Moreover, if the I-V curve plotted using the ln-ln scale in the HRS exhibits a complicated shape, the conduction mechanism is dominated by Space-Charge-Limited Conduction (SCLC). In the low-voltage region, the slope of the HRS is approximately 1. As bias voltage increases, the slope increases to 2 in the square region. When the bias voltage further increases, the current in the HRS increases rapidly and the slope in the high-voltage region is much sharper [15]. For the Al/Ga_2_O_3_/ITO/glass device, the slope in the HRS region changed from 1.01 to 1.91 to 3.1 with increasing voltage, which can be explained by SCLC, as shown in Figure 4a. For the Al/GO/Ga_2_O_3_/ITO/glass device, slopes of 1.05, 2.2, and 5.6 were obtained in low-voltage, square, and high-voltage regions (Figure 6d), indicating that the conduction mechanism follows SCLC. Furthermore, the results for the HRS were fitted to Schottky emission (ln(I)∝V^1/2^) and the Poole–Frenkel effect (ln(I/V) ∝V^1/2^), as shown in Figure 6c,d. The results were not fitted well by these two mechanisms.

Figure 7 shows the X-ray photoelectron spectroscopy (XPS) results for the surfaces of the Ga_2_O_3_ film and GO/Ga_2_O_3_ film. The O1s peak can be deconvoluted into two peaks at 531.3 ± 0.4 eV and 532.1 ± 0.2 eV through Gaussian fitting. The peak at 531.3 ± 0.4 eV is assigned to the well-bonded oxygen lattice bond with the metal cation, and the peak at 532.1 ± 0.2 eV is assigned to the oxygen vacancy bond. Thus, the area ratio of the oxygen vacancy-related peak to the entire O1s peak represents the relative quantity of oxygen. When oxygen content in the Ga_2_O_3_ film was increased from 27% to 77%, the area ratio of the oxygen vacancy-related peak to the entire O1s peak decreased from 37% to 29%. This result indicates that the number of oxygen vacancies in the GO/Ga_2_O_3_ film decreased with increasing oxygen content. Huang et al. reported that the amount of filament paths is proportional to the density of oxygen vacancies [16]. The binding energy of GO is lower than that of Ga–O, so oxygen ions can migrate more easily in GO, which implies that it helps oxygen migration during the SET/RESET process. Therefore, its set voltage is much lower (−1 V), it is more stable than a single layer, and its endurance is greater than that of the single layer.

Figure 8a shows the endurance tests for DC resistive switching. The current values were extracted at −0.1 V in the test. The Al/Ga_2_O_3_/ITO structure can maintain the ON/OFF ratio at around 10^4^ only after 28 DC switching cycles. This result demonstrates that the Al/Ga_2_O_3_/ITO structure was not stable for resistive switching. However, the Al/GO/Ga_2_O_3_/ITO structure can maintain the ON/OFF ratio at around 10^3^ after 126 DC switching cycles. This result demonstrates that the GO insertion layer played a key role in this improvement in the endurance test. Figure 8b shows the retention performance for the HRS and LRS, where no considerable degradation in resistance was observed after 10^4^ s at room temperature, indicating the device’s stability and nonvolatile storage properties.

Based on the filament theory, non-uniformity is an essential issue for RRAM devices due to the random conformation and rupture that occurs during set and reset processes, respectively. The two parameters generally observed in resistive switching characteristics to quantify non-uniformity are cycle-to-cycle uniformity and device-to-device uniformity.

Figure 9 shows the current distribution of the HRS and LRS. The current values in both states were measured at a read voltage of −0.1 volts and a compliance current of 100 mA. The coefficient of variation (CV) was used to evaluate the probability distribution and is defined as
$$CV=\frac{\sigma }{\left|\mu \right|}\times 100\%$$
where σ is the standard deviation and μ is the mean value [17]. Figure 9a shows that the result of cycle-to-cycle variation for the Al/Ga_2_O_3_/ITO device in the LRS at the average value was 3.61 × 10^−3^, and the coefficient of variation was about 20%. In the HRS at the average value, cycle-to-cycle variation was 7.21 × 10^−7^, and the coefficient of variation was about 83%. The HRS was not uniform during resistive switching. The result of cycle-to-cycle variation for the Al/GO/Ga_2_O_3_/ITO device in the LRS at the average value was 3.04 × 10^−3^, and the coefficient of variation was about 11%. In the HRS at the average value, cycle-to-cycle variation was 3.31 × 10^−6^, and the coefficient of variation was about 49%. Figure 9b shows that V_SET_ has about a 19% probability of being at −1.66 V, and V_RESET_ has about a 39% probability of being at 2.97 V for Ga_2_O_3_ devices. The mean and standard deviation of V_SET_/V_RESET_ for the GO/Ga_2_O_3_ devices are −1.04 V/17% and 2.59 V/30%, respectively. The distribution of switching voltages V_SET_ and V_RESET_ is very concentrated, highlighting the good uniformity.

Multi-level cell (MLC) storage is the simplest technology for accommodating more information at low cost in high-density storage applications. RRAM has particularly attracted lots of attention owing to its simplicity in obtaining multi-state characteristics. In the bilayer structure, the switching layer can be distinguished into a vacancy-rich region and vacancy-deficient region owing to the intrinsic nature of defect generation. The filament can be well controlled in the vacancy-poor region by modulating current compliance, which further obtains the multiple LRS to achieve MLC storage capability [18,19]. Figure 10a shows the I-V characteristics of the Al/GO/Ga_2_O_3_/ITO device at compliance currents of 100 mA, 10 mA, and 1 mA. Figure 10b shows the endurance characteristics of the Al/GO/Ga_2_O_3_/ITO device at 100 mA, 10 mA, and 1 mA current compliance. The ON/OFF ratio can be maintained at over 10^3^ after 126 DC cycles at a compliance current of 100 mA. The ON/OFF ratio can be maintained at over 10^3^ after 161 DC cycles at a compliance current of 10 mA and after 178 DC cycles at a compliance current of 1 mA. Figure 10c shows the retention characteristics of the bilayer device at 100 mA, 10 mA, and 1 mA current compliance. The result exhibits the possibility of 2-bit storage, including three LRSs and one HRS [20]. Figure 10d shows the current distribution of the HRS and LRS, which was used to assess device-to-device uniformity. Five different devices from the sample were randomly selected to conduct the DC I-V measurement for five cycles. Voltage sweeping from −4 V to 4 V was applied on all devices. For this analysis, the current values were also extracted at −0.1 volts. The results of device-to-device variation for GO/Ga_2_O_3_ at 100 mA, 10 mA, and 1 mA are also presented. Because of the GO layer, the GO/Ga_2_O_3_ device had excellent device-to-device uniformity.

Results for the cycle-to-cycle variation of the Al/GO/Ga_2_O_3_/ITO device at compliance currents of 100 mA, 10 mA, and 1 mA are also shown in Figure 11. The coefficient of variation values for the GO/Ga_2_O_3_ device in the LRS and HRS were about 11% and 49% at 100 mA. The coefficient of variation values for the GO/Ga_2_O_3_ device in the LRS and HRS were about 12% and 50% at 10 mA. The coefficient of variation values for the GO/Ga_2_O_3_ in the LRS and HRS were about 7% and 39% at 1 mA. Compared to the single-layer structure, the bilayer structure had better cycle-to-cycle uniformity because of the presence of the GO insertion layer.

Figure 12a plots the dependencies between the pattern sizes and resistive states of the Al/GO/Ga_2_O_3_/ITO/glass device at a current compliance of 10 mA. The resistance values were extracted at −0.1 V. This result demonstrates that both states were insensitive to pattern sizes. Thus, the filament theory appropriately explains the operation of resistive switching for this structure. Figure 12b shows the relationship between both states and temperatures at 10 mA current compliance. This result was characterized by an Agilent B1500A parameter analyzer, and the sample was heated to temperatures ranging from 300 K to 500 K. In this measurement, the resistance values were also extracted at −0.1 volts under different temperatures. The resistance of the HRS decreased with increasing temperature and exhibited semiconductor-like behavior. On the other hand, the current of the LRS was maintained at a fixed value under different temperatures. It is worth noting that the memory window was still maintained at around 10^2^ under the high temperature of 500 K, which could be attributed to the insertion of the GO layer.

Filament theory is the most widely used theory to elucidate the resistive switching mechanism of RRAM devices. It means the switching behavior relies on the conformation and rupture of the conductive filament (CF) in the switching layer [21,22]. In order to explain the resistive switching mechanism, filament models of the Al/Ga_2_O_3_/ITO/glass device and the Al/GO/Ga_2_O_3_/ITO/glass device are shown in Figure 13 and Figure 14, respectively. In the initial state, the Al/Ga_2_O_3_/ITO/glass and the Al/GO/Ga_2_O_3_/ITO/glass devices are not subjected to an applied bias, indicating an HRS state, as shown in Figure 13 and Figure 14.

Figure 13 shows that negative bias is applied on the Al electrode, and the oxygen ions will move from the Ga_2_O_3_ thin film to the ITO electrode under the action of the electric field, leaving oxygen vacancies to generate a conducting path. When the voltage reached V_set_, a conductive channel was formed by oxygen vacancies between the Ga_2_O_3_ thin films and the ITO electrode. Conversely, Figure 13 shows the top Al electrode was positively biased, and the O^2−^ in the Ga_2_O_3_ film migrated toward the positive Al electrode. The oxygen vacancies gradually decreased, the voltage reached V_reset_, and the conductive channel formed by oxygen vacancies between the top and bottom electrodes was broken, which describes the oxidation–reduction (redox) reaction. The migration of oxygen ions played an important role during resistive switching and resulted in filament formation and rupture. The ability to store oxygen ions at the anode substantially affected the resistance state’s stabilization.

As Figure 14 shows, when a negative bias is applied on the TE, the oxygen ion in the oxide moves toward the TE and leaves an oxygen vacancy behind, and the oxygen vacancy is arranged to form a conductive filament (this is the SET process). For the Al/GO/Ga_2_O_3_/ITO devices, this occurred due to the insertion of a GO layer between the Al and Ga_2_O_3_. On the contrary, when the top Al electrode was positively biased, O^2−^ moved in the opposite direction. At the same time, conduction filament ruptured due to the reverse redox reaction, as shown in Figure 14. The formation and rupture of the conduction filament had a certain randomness, which may lead to a large fluctuation in operating parameters (Figure 5a). The insertion of a GO layer would reduce this randomness. GO film can act as an oxygen ion reservoir. Compared with the Al/Ga_2_O_3_/ITO layer, the layered GO provided enhanced O^2−^ storage. Without external electrical stimuli, O^2−^ cannot easily migrate into or out of the GO substrate.

## 3. Conclusions

In this study, the Al/GO/Ga_2_O_3_/ITO/glass RRAM structure was demonstrated. Compared with the Ga_2_O_3_ monolayer structure, bilayer RRAM had better electrical properties, such as lower forming and SET/RESET voltages, as well as better endurance. Based on the filament theory, the formation of conductive filament, which is composed of oxygen vacancies, can affect resistive switching. Because of the GO layer, the switching layer could be distinguished into a vacancy-rich region (GO) and vacancy-deficient region (Ga_2_O_3_), which helped us to control the conformation and rupture of CF in order to optimize the endurance characteristics. Endurance was improved to more than 100 cycles, which was five times higher. Moreover, the I-V characteristics and endurance tests (modulated by decreasing the compliance current to 100 mA, 10 mA, and 1 mA to achieve the multi-level cell (MLC) technology) were also described. It is an excellent technology for accommodating a large amount of data at low cost. Furthermore, the bilayer RRAM had stable retention for more than 10^4^ s without significant degradation in all states. Finally, the filament models were also described to explain the conformation and rupture in the resistive switching layers of both structures.

## 4. Materials and Methods

A schematic of the Al/GO/Ga_2_O_3_/ITO structure is shown in Figure 15a. The RRAM device was fabricated on a commercial ITO/glass substrate. We used the ITO layer as the bottom electrode (BE). RF sputtering was operated at a power of 90 W and a working pressure of 6mTorr under Ar/O_2_ flow of 25/25sccm for 10 min to deposit a 15 nm Ga_2_O_3_ switching layer. Graphite and potassium permanganate were slowly added to a mixture of sulfuric and phosphoric acids at room temperature and stirred for 24 h. The mixture was then treated with water and hydrochloric acid until the pH was neutral. Finally, graphene was obtained in two centrifugation steps at 6000× *g* rpm for 3 h. A graphene oxide solution was obtained by dissolving graphene in water. The graphene oxide solution was then spin-coated onto Ga_2_O_3_/ITO/glass samples at 700/1000 rpm for 20/40 s and vacuum baked at 60 °C for 10 min to obtain graphene oxide films [23,24]. Finally, the 66 nm Al top electrode (TE) patterned by a shadow mask with 0.25 mm^2^ holes was deposited on the GO/Ga_2_O_3_/ITO/glass sample using DC sputtering. Furthermore, the Al/Ga_2_O_3_/ITO/glass structure was also fabricated for comparison. The electrical measurements were carried out using an Agilent B1500A semiconductor parameter analyzer. Bias was applied on the TE and the BE was grounded. Figure 15b depicts a TEM image of the device’s cross-section. It is worth noting that a sub-AlO_x_ interfacial layer was formed between the Al metal layer and GO layer because the Al scavenged the oxygen ions from the GO layer. The thickness values of the GO layer and AlO_x_ layer were ~4 nm and ~4 nm, respectively.

## Figures and Tables

**Figure 1 nanomaterials-13-01851-f001:**
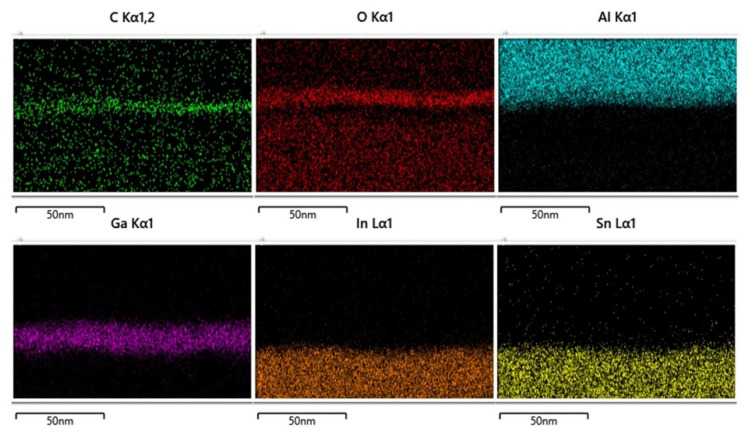
EDS mapping of the cross-section profile of the device.

**Figure 2 nanomaterials-13-01851-f002:**
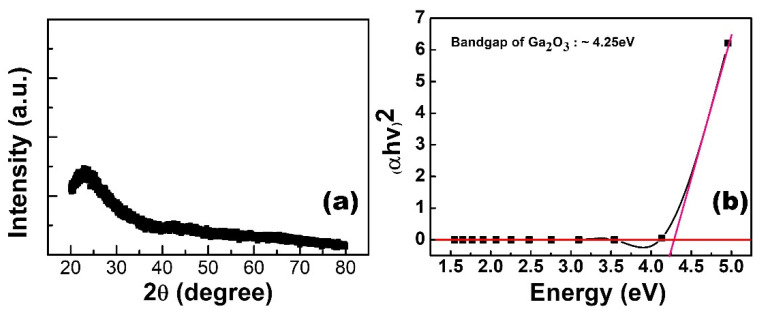
(**a**) The XRD result of Ga_2_O_3_ thin film on a glass substrate. (**b**) The bandgap of Ga_2_O_3_ as measured by a spectrophotometer.

**Figure 3 nanomaterials-13-01851-f003:**
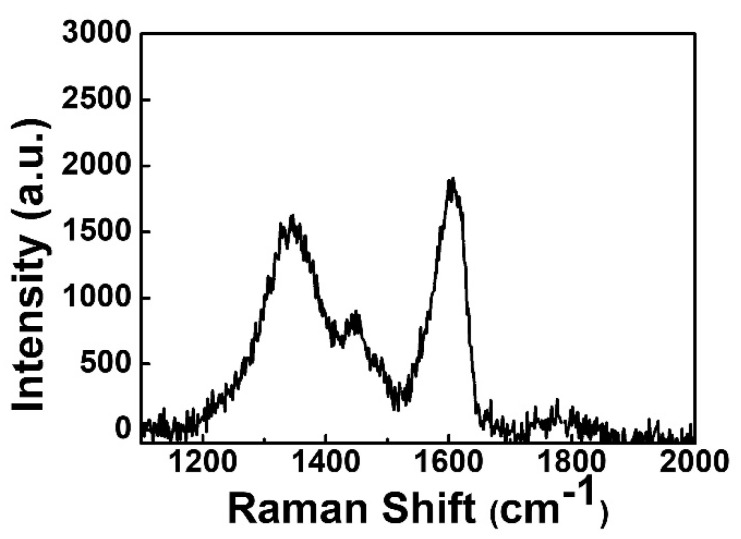
Raman spectroscopy of GO thin film.

**Figure 4 nanomaterials-13-01851-f004:**
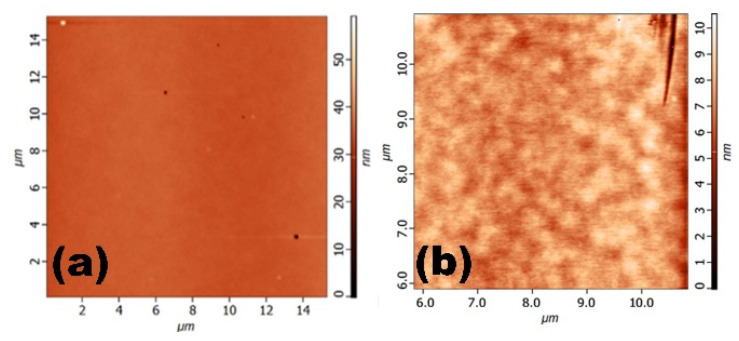
AFM image of the (**a**) Ga_2_O_3_ and (**b**) GO/Ga_2_O_3_ thin films.

**Figure 5 nanomaterials-13-01851-f005:**
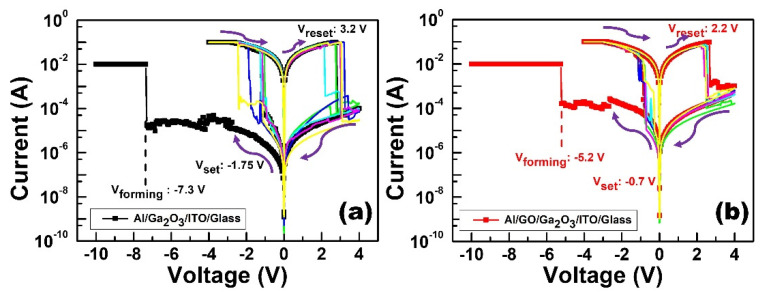
I-V switching cycle curves of the (**a**) Al/Ga_2_O_3_/ITO/glass and (**b**) Al/GO/Ga_2_O_3_/ITO/glass RRAM devices.

**Figure 6 nanomaterials-13-01851-f006:**
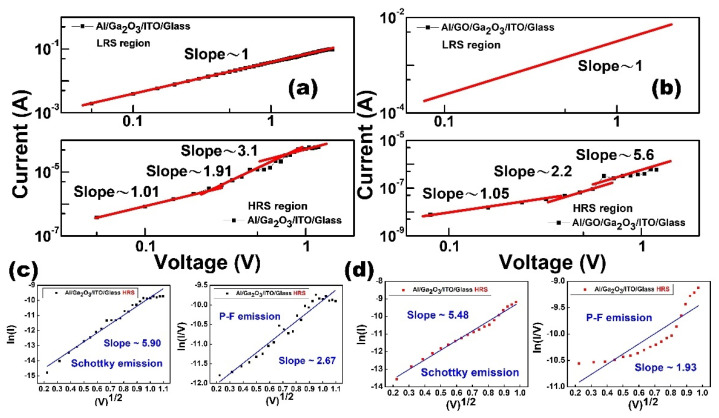
Double logarithmic plot fitting of the results of the HRS and LRS for the (**a**) Ga_2_O_3_ and (**b**) GO/Ga_2_O_3_ devices. The fitting results of the HRS used Schottky emission and the Poole–Frenkel effect for (**c**) Ga_2_O_3_ and (**d**) GO/Ga_2_O_3_ devices.

**Figure 7 nanomaterials-13-01851-f007:**
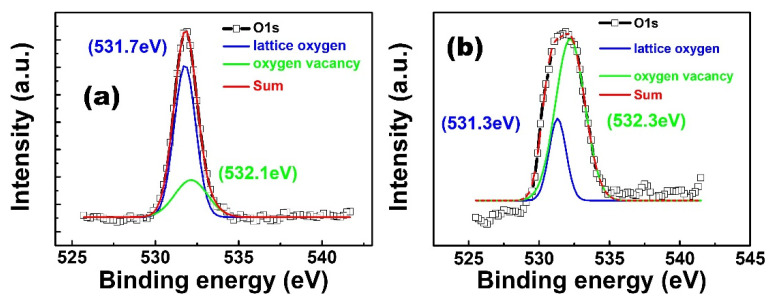
XPS O1s spectra of (**a**) Ga_2_O_3_ film and (**b**) GO/Ga_2_O_3_ thin film.

**Figure 8 nanomaterials-13-01851-f008:**
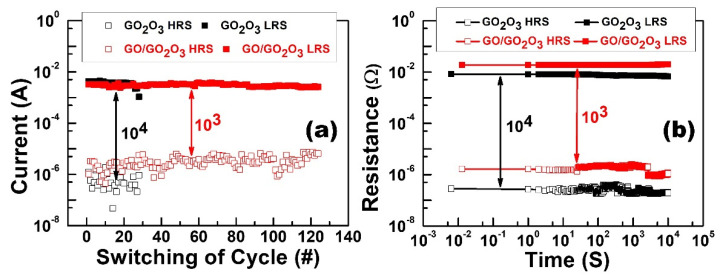
(**a**) Endurance and (**b**) retention (@T = 300 K) characteristics of Ga_2_O_3_ and GO/Ga_2_O_3_ devices in the HRS and LRS.

**Figure 9 nanomaterials-13-01851-f009:**
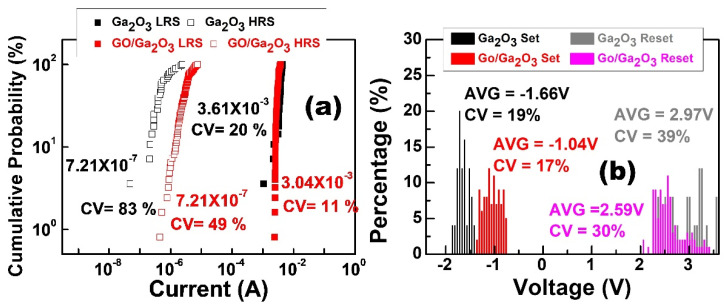
(**a**) Cumulative probability and (**b**) operation voltage distributions of the HRS and LRS in terms of cycle-to-cycle variability in the Ga_2_O_3_ and GO/Ga_2_O_3_ devices.

**Figure 10 nanomaterials-13-01851-f010:**
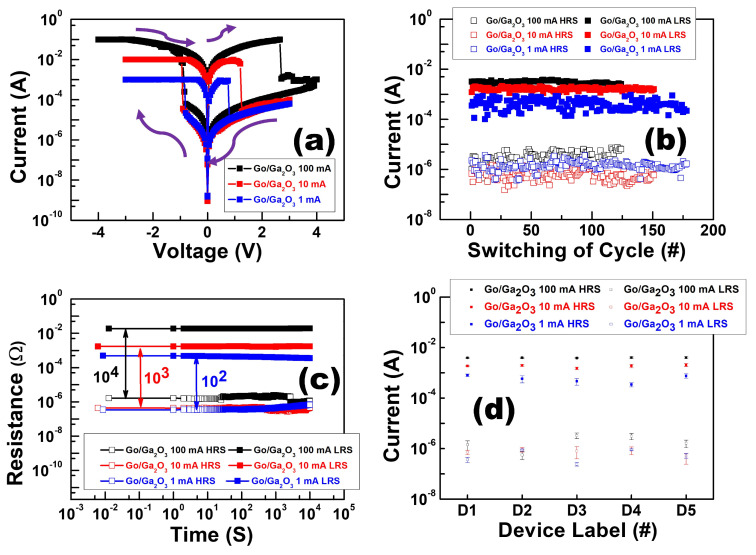
(**a**) I-V characteristics, (**b**) endurance tests, (**c**) retention characteristics, and (**d**) device-to-device variation for Al/GO/Ga_2_O_3_/ITO under 100 mA, 10 mA, and 1 mA current compliance.

**Figure 11 nanomaterials-13-01851-f011:**
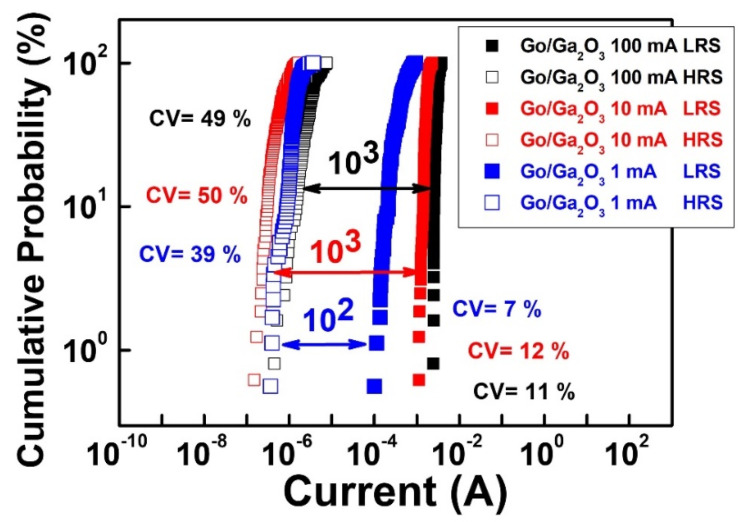
Cumulative cycle-to-cycle probability of the HRS and LRS in the GO/Ga_2_O_3_ device at compliance currents of 100 mA, 10 mA, and 1 mA.

**Figure 12 nanomaterials-13-01851-f012:**
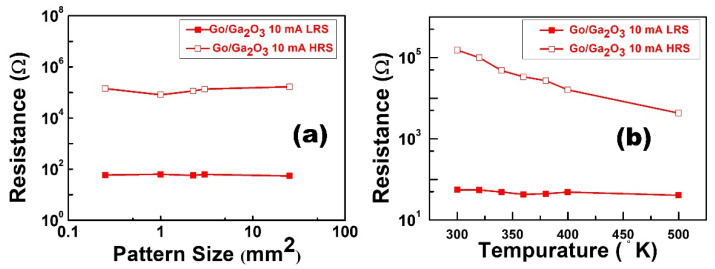
(**a**) Pattern size-dependent measurements and (**b**) temperature-dependent resistance measurements between 300 K and 500 K for the Al/GO/Ga_2_O_3_/ITO/glass device at a current compliance of 10 mA.

**Figure 13 nanomaterials-13-01851-f013:**
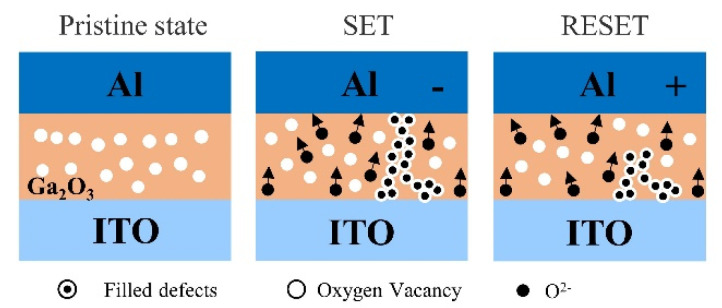
Filament models of the Al/Ga_2_O_3_/ITO/glass device.

**Figure 14 nanomaterials-13-01851-f014:**
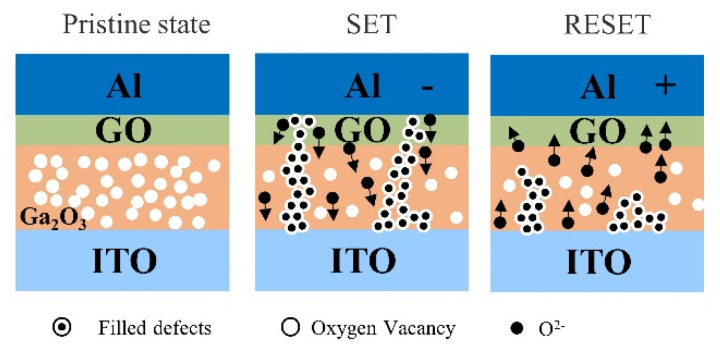
Filament models of the Al/GO/Ga_2_O_3_/ITO/glass device.

**Figure 15 nanomaterials-13-01851-f015:**
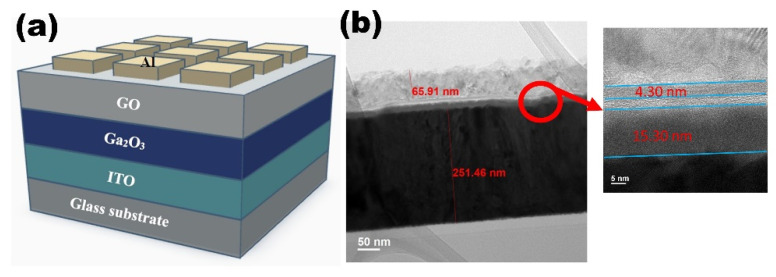
(**a**) The schematic and (**b**) TEM cross-section images of Al/GO/Ga_2_O_3_/ITO/glass RRAM.

## Data Availability

Not applicable.

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
