# Peer review of "Multi-Level Resistive Al/Ga2O3/ITO Switching Devices with Interlayers of Graphene Oxide for Neuromorphic Computing"

_nanomaterials, 2023, doi:10.3390/nano13121851_

Round 1
Reviewer 1 Report
The article "Switching Performance Enhancement in Gallium Oxide-based Multilevel RRAM Devices using Graphene Oxide Insertion Layer" is devoted to the study of resistive switching in Al/Ga2O3/ITO and Al/GO/Ga2O3/ITO structures. The article presents the electrophysical characteristics of the structures, analysis the conduction mechanisms and proposes a filamentary model of resistive switching. However, there are some revisions, after the correction of which the article can be published in the journal Nanomaterials.
1) All figures must be enlarged.
2) In some figures (Fig. 3b, 5b, 6a, 7c, 8) the legend is outside the plot area. It will look more accurate if placed inside.
3) On Fig.7a I-V characteristics are shown only for the structure with GO. It should be better to show such graph for the another structure to the comparison of multibit operation possibility.
4) There are only 100-200 switching shown, but for the modern RRAM devices the usual value is 10^6 and higher. Is it possible to measure more switching cycles or is it a device limitation? If yes, please explain why.
5) One of the main question is the switching mechanism. It is well-known, that the gallium oxide is the material in which oxygen vacancies play the main role on the parameters of this material - electrical, optical etc. But, in model (and on corresponding Fig. 10) oxygen vacancies in Ga2O3 are not taken into account. Moreover, there is no explanation about the positive role of GO. This fact should be discussed separately in the manuscript.
Author Response
1) All figures must be enlarged.
Reply:Thank you for the suggestions. In the revision, the resolution of all figures has been improved.
2) In some figures (Fig. 3b, 5b, 6a, 7c, 8) the legend is outside the plot area. It will look more accurate if placed inside.
Reply:Thank you for the suggestions. In the revision, the resolution of all figures has been improved.
3) On Fig.7a I-V characteristics are shown only for the structure with GO. It should be better to show such graph for the another structure to the comparison of multibit operation possibility.
Reply:Thank you for your comments. We compare Al/Ga2O3/ITO and Al/GO/Ga2O3/ITO device at compliance currents of 100 mA, 10 mA and 1 mA, respectively. Since the IV performance of Al/Ga2O3/ITO device is not as good as expected, we will discuss Al/GO/Ga2O3/ITO device components in this paper in Fig. 1.
Fig. 1. The I-V characteristics (a) Al/Ga2O3/ITO and (b) Al/GO/Ga2O3/ITO under the current compliance of 100 mA, 10mA and 1mA.
4) There are only 100-200 switching shown, but for the modern RRAM devices the usual value is 10^6 and higher. Is it possible to measure more switching cycles or is it a device limitation? If yes, please explain why.
Reply:Thank you for your comments. The 100-200 sample tests mentioned in the manuscript refer to cycle to cycle tests, not device to device. Due to the design of the photomasks, there are only a maximum of 56 devices on a single specimen. Below we provide the statistics of 56 devices. As shown in Fig. 2, the percentage of the device that could work for at least 56 cycles of Ga2O3 and GO/Ga2O3 devices are 38%, and 64%, respectively.
Fig. 2. Operation voltage distributions of (a) Ga2O3 and (b) GO/Ga2O3 RRAM devices for cycle-to-cycle.
5) One of the main question is the switching mechanism. It is well-known, that the gallium oxide is the material in which oxygen vacancies play the main role on the parameters of this material - electrical, optical etc. But, in model (and on corresponding Fig. 10) oxygen vacancies in Ga2O3 are not taken into account. Moreover, there is no explanation about the positive role of GO. This fact should be discussed separately in the manuscript.
Reply:Thank you for your valuable suggestions. After careful consideration, we have added XPS results this paragraph before the discussion of switching mechanisms. In the revision, we have revised the following statements (Page 4): “Figure 3 shows the X-ray photoelectron spectroscopy (XPS) results on the surface of Ga2O3 film and GO/Ga2O3 film. The O 1 s peak can be deconvoluted into two peaks at 531.3± 0.4 eV, and 532.1± 0.2 eV through Gaussian fitting. The peak at 531.3± 0.4 eV is as-signed to the well-bonded oxygen lattice bond with the metal cation, and the peak at 532.1± 0.2 eV is assigned to the oxygen vacancy bond. Thus, the area ratio of the oxygen vacancy related peak to the entire O1s peak represents the relative quantity of oxygen. When the oxygen content in the Ga2O3 film was increased from 27% to 77%, the oxygen vacancy-related peak area ratio decreased from 37% to 29%. This result indicates that the number of oxygen vacancies in the GO/Ga2 O3 film decreased with increasing oxygen con-tent. Huang et al. reported that the amount of filament paths is proportional to the density of oxygen vacancies [11]. The binding energy of GO is lower than that of Ga–O, so oxy-gen ions can migrate more easily in GO, which implies that it helps oxygen migration during the set/reset process. Therefore, its set voltage is much lower (-1 V), it is more stable than a single layer, an d its endurance is greater than that of the singlelayer.”
Figure 3. XPS O 1s spectra of (a) Ga2 O3 film and (b) GO/Ga2 O3 thin film
- 1. H. W. Huang, C. F. Kang, F. Lai, J. H. He, S. J. Lin, and Y. L. Chueh, “Stability scheme of ZnO-thin film resistive switching
memory: influence of defects by controllable oxygen pressure ratio,” Nanoscale Research Letters, vol. 8, p. 483, 2013

Reviewer 2 Report
The diagrams are too small to be legible.
The English can be improved, although the mistakes do not interfere with understanding the material. A few English problem examples are:
line 20 in abstract: "highly-profile"
line 21 in abstract: "already be widely used"
line 55 "few group have"
Author Response
Review Report Form 2
Open Review
( )I would not like to sign my review report
(x) I would like to sign my review report
Quality of English Language
(x) I am not qualified to assess the quality of English in this paper
( ) English very difficult to understand/incomprehensible
( ) Extensive editing of English language required
( ) Moderate editing of English language
( ) Minor editing of English language required
( ) English language fine. No issues detected
Yes Can be improved Must be improved Not applicable
Does the introduction provide sufficient
background and include all relevant references? (x) ( ) ( ) ( )
Are all the cited references relevant to the research? (x) ( ) ( ) ( )
Is the research design appropriate? (x) ( ) ( ) ( )
Are the methods adequately described? (x) ( ) ( ) ( )
Are the results clearly presented? (x) ( ) ( ) ( )
Are the conclusions supported by the results? (x) ( ) ( ) ( )
Comments and Suggestions for Authors |
The diagrams are too small to be legible. |
Comments on the Quality of English Language |
The English can be improved, although the mistakes do not interfere with understanding the material. A few English problem examples are: |
- line 20 in abstract: "highly-profile"
Reply:We apologize for the typos. In the revision, we have revised the following statements (Page 1 line 20): “Among numerous semiconductor materials, gallium oxide, as the fourth-generation semiconductor material, due to its excellent material properties, and applied in the field of optoelectronics high-power, resistive switching device and so on, due to its wide bandgap and transparent properties.”
- line 21 in abstract: "already be widely used"
Reply:We apologize for the typos. In the revision, we have revised the following statements (Page 1 line 21): “Among numerous semiconductor materials, gallium oxide, as the fourth-generation semiconductor material, due to its excellent material properties, and applied in the field of optoelectronics high-power, resistive switching device and so on, due to its wide bandgap and transparent properties.”
- line 55 "few group have"
Reply:We apologize for the typos. In the revision, we have revised the following statements (Page 2 line 55): “However, some teams have studied Ga2O3 material for memory applications.”
Submission Date 05 May 2023
Date of this review 15 May 2023 22:20:28

Reviewer 3 Report
There are a number of comments to the work that need to be corrected.
1. Small drawings with low DPI.
2. Misprint in the numbering of figures (Fig. 6 instead of Fig. 4 on the 3rd page).
3. The text does not explain what GO is.
4. Arrows should be added to Figure 3b showing the direction of voltage change in the corresponding parts of the I-V curves.
5. In Figure 3,b, I would like to see not only the first I-V curve, but also several subsequent ones to confirm the measured values.
6. In fig. 4 claims a graph with logarithmic scales, while numerical values correspond to linear scales.
7. For clarity, it is necessary in fig. 4 note which charts correspond to HRS and which ones - to LRS.
8. All reproducibility studies have a different number of cycles, what is the reason for this?
There are a number of general remarks about the work.
Author Response
Review Report Form 3
Open Review
( )I would not like to sign my review report
(x) I would like to sign my review report
Quality of English Language
(x) I am not qualified to assess the quality of English in this paper
( ) English very difficult to understand/incomprehensible
( ) Extensive editing of English language required
( ) Moderate editing of English language
( ) Minor editing of English language required
( ) English language fine. No issues detected
Yes Can be improved Must be improved Not applicable
Does the introduction provide sufficient
background and include all relevant references? (x) ( ) ( ) ( )
Are all the cited references relevant to the research? ( ) (x) ( ) ( )
Is the research design appropriate? ( ) (x) ( ) ( )
Are the methods adequately described? (x) ( ) ( ) ( )
Are the results clearly presented? ( ) ( ) (x) ( )
Are the conclusions supported by the results? (x) ( ) ( ) ( )
Comments and Suggestions for Authors |
The diagrams are too small to be legible. |
Comments on the Quality of English Language |
There are a number of comments to the work that need to be corrected. |
- Small drawings with low DPI. 1.
Reply:Thank you for the suggestions. In the revision, the resolution of all figures has been improved.
- Misprint in the numbering of figures (Fig. 6 instead of Fig. 4 on the 3rd page).
Reply:Thank you for your comments. We have carefully revised the manuscript and all figures, considered the reviewers' comments and made corrections.
- The text does not explain what GO is.
Reply:Thank you for your comments. We have carefully revised the sentence and corrected it to “In this study, we have successfully demonstrated the Al/Graphene oxide (GO)/Ga2O3/ITO RRAM to achieve the possibility of two-bit storage.”
- Arrows should be added to Figure 3b showing the direction of voltage change in the corresponding parts of the I-V curves.
Reply:Thank you for your comments. In the revision, we have revised Fig. 3 and the following statements (Page 3): “The ON/OFF ratios of the Al/Ga2O3/ITO/glass and Al/GO/Ga2O3/ITO/glass RRAM devices remained stable for more than 20 cycles.”
Figure 3. I-V switching cycles curves of the (a) Al/Ga2O3/ITO/glass and (b) Al/GO/Ga2O3/ITO/glass devices RRAMs.
- In Figure 3,b, I would like to see not only the first I-V curve, but also several subsequent ones to confirm the measured values.
Reply:Thank you for your comments. In the revision, we have revised Fig. 3 and the following statements (Page 3): “The ON/OFF ratios of the Al/Ga2O3/ITO/glass and Al/GO/Ga2O3/ITO/glass RRAM devices remained stable for more than 20 cycles.”
Figure 3. I-V switching cycles curves of the (a) Al/Ga2O3/ITO/glass and (b) Al/GO/Ga2O3/ITO/glass devices RRAMs.
- In fig. 4 claims a graph with logarithmic scales, while numerical values correspond to linear scales.
Reply:Thank you for your comments. We have replotted the figure 4.
Figure 4. Double logarithmic plot fitting results of the HRS and LRS for the (a) Ga2O3 and (b) GO/Ga2O3 devices. The fitting results of HRS using Schottky emission and Poole-Frenkel (c) Ga2O3 and (d) GO/Ga2O3 devices.
- For clarity, it is necessary in fig. 4 note which charts correspond to HRS and which ones - to LRS.
Reply:Thank you for your comments. We have replotted the figure 4.
Figure 4. Double logarithmic plot fitting results of the HRS and LRS for the (a) Ga2O3 and (b) GO/Ga2O3 devices. The fitting results of HRS using Schottky emission and Poole-Frenkel (c) Ga2O3 and (d) GO/Ga2O3 devices.
- All reproducibility studies have a different number of cycles, what is the reason for this? There are a number of general remarks about the work.
Reply:Thank you for your comments. The coefficient of variation (CV), defined as the ratio of standard the deviation to average value (AVG), is used to evaluate distribution. The distribution of the HRS current was larger than the LRS current because the different filament sizes cannot rupture completely through the same bias.
Submission Date 05 May 2023
Date of this review 10 May 2023 10:01:30

Round 2
Reviewer 1 Report
Article could be published in the present form.